# Clinical Significance of Circulating Tumor Cells in Epithelial Appendiceal Neoplasms with Peritoneal Metastases

**DOI:** 10.3390/cancers16132441

**Published:** 2024-07-02

**Authors:** Petter Frühling, Louice Moberg, Lana Ghanipour, Helgi Birgisson, Wilhelm Graf, Christer Ericsson, Peter H. Cashin

**Affiliations:** 1Uppsala Sweden and Department of Surgery, Institution of Surgical Sciences, Uppsala University, Akademiska Sjukhuset, 751 85 Uppsala, Sweden; louice.moberg@regionkalmar.se (L.M.); lana.ghanipour@uu.se (L.G.); helgi.birgisson@uu.se (H.B.); wilhelm.graf@uu.se (W.G.); peter.cashin@uu.se (P.H.C.); 2iCellate Medical AB, KI Science Park, Industrivägen 1, 171 48 Solna, Sweden; christer.ericsson@icellate.se; 3Department of Microbiology, Tumor and Cell Biology (MTC), Biomedicum 8 C, Karolinska Institute, 171 77 Stockholm, Sweden

**Keywords:** appendiceal neoplasm, pseudomyxoma peritonei, circulating tumor cells, cytoreductive surgery, hyperthermic intraperitoneal chemotherapy

## Abstract

**Simple Summary:**

This study aimed to assess the prognostic role of circulating tumor cells (CTCs) in patients with epithelial appendiceal neoplasms with peritoneal metastases. The presence of CTCs may be used for the early detection of invasive cancer in this rare diagnosis. Our study is the first study to assess the potential value of CTCs in this specific group of patients.

**Abstract:**

Appendiceal tumors are uncommon and, at times, discovered incidentally during histological examination. The histopathological classification of the disease is complex and has generated some controversy. The analysis of circulating tumor cells can be used for the early detection of metastatic potential. The aim of the present study was to examine the prognostic value of circulating tumor cells in patients with appendiceal tumors and peritoneal metastases. To our knowledge, this is the first study to examine CTCs in appendiceal tumors. We performed a prospective cohort study of consecutive patients treated with cytoreductive surgery and hyperthermic intraperitoneal chemotherapy between 2015 and 2019 at a HIPEC referral center. In total, 31 patients were included in the analysis, and circulating tumor cells were detected in 15 patients (48%). CTC positivity was not associated with overall or recurrence-free survival, nor was it correlated with PCI score or histopathological grading. Surprisingly, however, CTCs were found in almost half the patients. The presence or quantities of these cells did not, on their own, predict systemic metastatic potential during the observed time, and they did not appear to significantly correlate with the oncological outcomes recorded.

## 1. Introduction

Appendiceal neoplasms are relatively rare and heterogenous, which, at times, are discovered incidentally during the pathological analysis of the appendix. In a large retrospective study, acute appendicitis was the most common presentation, and neoplasms were found in less than one percent of the cases [1]. Tumors of the appendix constitute less than 0.5 percent of all intestinal tumors [2], and are often treated as if they were a colorectal cancer [3]. Because of the scarcity of clinical and pre-clinical research that focuses specifically on appendix neoplasms, our knowledge of this rare disease is limited. The classification of neoplasms of the appendix is complex [4] and they are broadly classified as epithelial (mucinous, non-mucinous, signet-ring cell tumors, and adenocarcinoma) and non-epithelial [e.g., neuroendocrine tumors]. In addition, goblet cell carcinoma constitutes an aggressive form, which shares characteristics of both groups [5]. The mucinous group of appendiceal tumors is highly heterogenous, ranging from simple mucoceles to a wide spread peritoneal growth known as pseudomyxoma peritonei (PMP) [6,7,8]. Appendiceal mucinous tumors are classified into low-grade appendiceal mucinous neoplasms (LAMNs) confined to the mucosa of the appendix, and high-grade mucinous adenocarcinoma, which is invasive and spread beyond the muscularis mucosa [9,10]. There is also an intermediate group known as high-grade appendiceal mucinous neoplasms (HAMNs), which are more rarely diagnosed than LAMNs and are characterized by high-grade dysplasia of the epithelium. For the mucinous appendiceal neoplasms with peritoneal disease, the histopathology of the peritoneal disease determines the prognosis more than the primary tumor. The following four categories are now used to classify the peritoneal disease—acellular mucin, mucinous carcinoma peritonei grade 1 (diffuse peritoneal adenomucinosis—DPAM), mucinous carcinoma peritonei grade 2 (peritoneal mucinous carcinoma—PMCA), and mucinous carcinoma peritonei grade 3 (PMCA with signet-ring cells). Non-mucinous tumors of the appendix include adenoma, adenocarcinoma, and signet-ring cell carcinoma [9].

Recent research suggests that appendiceal neoplasms have key molecular drivers different from colorectal cancer and represent a distinct clinico-molecular entity that does not resemble right-sided colorectal cancer, or any of the four consensus molecular subtypes [11,12]. The presence of peritoneal metastases of grades 1 and 2 imply poor prognosis and were, in the past, considered an incurable condition [13], but today the treatment of appendiceal tumors with the presence of peritoneal metastases often involves the combination of cytoreductive surgery (CRS) and hyperthermic intraperitoneal chemotherapy (HIPEC). The mechanisms of cancer metastasis is a multi-step process, which is still only partly understood [14,15]. Circulating tumor cells may play an important role in metastatic spread and disease progression and might be used as a prognostic biomarkers in gastrointestinal cancers [16]. Evidently, there is a great need for reliable biomarkers that could guide treatment decisions. To our knowledge, this is the first study to investigate the clinical significance of circulating tumor cells (CTCs) in patients with appendiceal tumors and peritoneal metastasis. 

The aim of this exploratory study was to examine the presence of CTCs and their influence on recurrence and overall survival in patients with appendiceal tumors with peritoneal metastases. 

## 2. Materials and Methods

### 2.1. Patients

This prospective study comprised consecutive patients with appendiceal neoplasms who underwent cytoreductive surgery and hyperthermic intraperitoneal chemotherapy, enrolled from November 2015 and September 2019 at Uppsala University Hospital. 

### 2.2. Work-Up of Patients and Follow-Up

All patients were considered to be candidates for CRS and HIPEC and were discussed in a tumor board meeting, which included HIPEC surgeons, radiologists, and oncologists. Patients were staged with contrast-enhanced computed tomography (CE-CT) of the abdomen and chest and, in some instances, with magnetic resonance imaging (MRI) and ^18^F-fluorodeoxyglucose (FDG)-positron emission tomography (FDG-PET/CT). Patients in which it was obvious at initial surgical exploration that adequate CRS (CCS 0-1) could not be obtained and/or palliative debulking was performed were classified as open-and-close procedures (CCS-3). Patients were followed-up in the outpatient clinic 4–6 weeks after surgery, and follow-up imaging was performed in accordance with the national guidelines for metastatic colorectal cancer [17].

### 2.3. Circulating Tumor Cell (CTC) Isolation and Enrichment

Circulating tumor cells (CTCs) are rare [18] but are relatively large cells [19]. To increase the probability of CTC isolation and detection, two blood samples were taken: the first sample, one to four days prior to surgery, and the second on the day of surgery. The mean volume of blood that was extracted was 10.7 mL, and CTC-positivity was defined as at least 1 CTC/8 mL of blood. After blood collection, the specimen was marked with a study number and sent within 48 h to iCellate Medical AB laboratory in Stockholm, Sweden. The specimens were analyzed anonymously, without any knowledge of the patient’s medical history or clinical outcome.

CTC detection and enumeration were performed using iCellate’s Medical AB detection technology, which follows the CellMate^®^ protocol. The CellMate^®^ system is a novel approach that uses a microfluidic device that identifies different cells based on their biomechanical properties, e.g., flow rate, surface interactions, elasticity, and plasticity. It can detect and enrich CTCs to facilitate scoring (Figure 1) [20,21].

### 2.4. Statistical Analysis

We reported the median and interquartile ranges (IQR) for ages at surgery. Categorical variables were compared with Fisher’s exact test and continuous variables with the Mann–Whitney U test. Overall survival was calculated from the date of surgery until death. Recurrence was deemed present if clearly visible on follow-up radiological imaging. Time to recurrence and overall survival were estimated with the Kaplan–Meier method, and groups were compared with the with the Cox-f test. Median follow-up time was calculated with the reversed Kaplan–Meier method. All statistical analyses were performed with SPSS version 28, and STATISTICA v14.1. Statistical significance was set at *p* < 0.05.

### 2.5. Permissions

All patients provided informed consent, and the study was approved by the Swedish Ethical Review Authority (Dnr. 2015/129). The study was registered online as follows: NCT04083547 (ClinicalTrials.gov).

## 3. Results

Thirty-one patients were included in the analysis (Figure 2).

Seventeen patients (53%) were men, and fourteen patients (47%) were women. The median age at surgery was 58 years (54–67 IQR) for the total cohort (Table 1). In 15 patients (48%), CTCs were identified and classified as CTC-positive. Sixteen patients (52%) were classified as CTC-negative since no CTCs were detected. Sixty-seven percent (n = 10) in the CTC-positive group belonged to Karnofsky grade 100, and 71% (n = 12) in the CTC-negative group. There were no differences between the CTC-positive and CTC-negative groups with regards to age, sex, BMI, or Karnofsky grade. 

### 3.1. Histopathology and Peritoneal Histopathology

Nine patients (60%) in the CTC-positive group and six patients (38%) in the CTC-negative group had low-grade appendiceal mucinous neoplasms (LAMN) (*p* = 0.376) (Table 1). Adenocarcinoma was diagnosed in 27% (n = 4) of patients in the CTC-positive group and 44% (n = 7) in the CTC-negative group (Table 1). Six patients (CTC-positive n = 3, CTC-negative n = 3) had disseminated peritoneal adenomucinosis (DPAM) and seven patients (CTC-positive n = 3, CTC-negative n = 4) peritoneal mucinous carcinomatosis (PMCA). A majority of patients (n = 10, 32%) had adenocarcinoma, and another two patients had adenocarcinoma with the presence of signet cells. Fifty-five percent (n = 17) of patients had mucinous carcinoma peritonei grade 2 (CTC-positive n = 7, CTC-negative n = 10). Twenty patients (64%) had high-grade tumors (CTC-positive n = 9, and CTC-negative n = 11).

#### 3.1.1. Surgical Results, Chemotherapy, and Complications

Surgical treatment and related characteristics are reported in Table 2. A majority of all patients (71%) underwent both CRS and HIPEC. Nine patients in the CTC-positive group and thirteen patients in the CTC-negative group had CRS together with HIPEC (Table 2). Cytoreductive surgery only was performed in two patients and debulking in three patients. Open-and-close procedures were performed in four (27%) patients, all of whom were CTC-positive. 

The median peritoneal cancer index (PCI) was 26 and 24 for the CTC-positive and CTC-negative groups, respectively (*p* = 0.880). There was no difference in the completeness of cytoreduction score (CCS) between the groups, and a CCS score of 0 was obtained in 48% of the cases. Six patients experienced a complication rated of Clavien–Dindo 3a or greater. The complications included wound dehiscence (n = 1), intra-abdominal bleeding (n = 1), bowel perforation (n = 1), multiorgan failure (n = 1), and pulmonary embolism (n = 1). Neoadjuvant chemotherapy was given to two patients that belonged to the CTC-positive group. Adjuvant chemotherapy was given to four patients in the CTC-positive group, and five patients in the CTC-negative group (*p* = 0.647).

#### 3.1.2. Time to Recurrence and Overall Survival

Median overall survival in the CTC-positive group was not reached, with a 5-year survival rate of 53%; in the CTC-negative group, median overall survival was 66 months (95% CI 34—not reached), with a 5-year survival rate of 55% (Cox’s F-test *p* = 0.355) (Figure 3a). There was no difference in time to recurrence or overall survival between the CTC groups (Figure 3a,b). Subgroup analyses were performed for CC0-1 and high-grade tumors: no differences were seen. 

In a multivariable Cox’s regression analysis, no association could be found between advanced age (above 70 years old), sex (men or women), whether CTCs could be detected in the bloodstream, or whether it was a low- or high-grade tumor and overall survival (Table 3). However, there was a strong correlation between a PCI score of 13 or above and overall survival (HR 5.21, 95% CI 1.16–23.38, *p* = 0.031). In terms of recurrence-free survival, a correlation was found with both a PCI score of 13 and above (HR 5.17, 95% CI 1.60–16.63, *p* = 0.006), and high-grade tumors (HR 3.97, 95% CI 1.18–13.40, *p* = 0.026). No association was found between CTC-positivity and earlier recurrence of the disease (Table 4).

## 4. Discussion

The presence of circulating tumor cells has been shown to be prognostic in metastatic breast, prostate, and colorectal cancer [22,23,24]. Furthermore, in a recent study by Nicolazzo et al., the presence of CTCs in a subgroup of patients with bladder cancer (HRT1 bladder cancer) was strongly correlated with a high risk of local recurrence and/or progression of the disease [25]. The present study is the first to assess the clinical significance of circulating tumor cells (CTCs) in patients with epithelial appendiceal neoplasms with peritoneal metastases. Appendiceal neoplasms are rare tumors, and their incidence appears to be increasing. In a recent population-based study, the age-standardized incidence of appendiceal mucinous tumors was estimated to be less than two cases per 1,000,000 person-years [26]. We performed a prospective cohort study of consecutive patients treated with cytoreductive surgery (CRS) and hyperthermic intraperitoneal chemotherapy (HIPEC) between 2015 and 2019 at a HIPEC referral center. In total, 31 patients were included in the analysis. Fifteen patients had CTCs present in their blood, and sixteen patients were CTC-negative. No differences were discerned between the groups (CTC-negative versus CTC-positive) in terms of the baseline characteristics of the study population (Table 1). 

CTCs may provide important information regarding the tumor biology and the metastatic process [27], since they constitute the smallest complete unit of metastasis, but are demanding to isolate and analyze. Consequently, CTC-based assays are only gradually starting to enter clinical practice. Ideally, representative CTC sampling and analyses should work as a complement to radiological imaging in clinical practice. Potential clinical applications include the early detection of cancer, surveillance for micro-metastatic disease, treatment selection, and response monitoring in patients with metastatic disease [28]. In a study by Busetto et al., it was found that CTCs may play a role in risk stratification, both for the recurrence and progression of the disease, for patients with high-risk non-muscle invasive bladder cancer [29]. In addition, in a recently published study by the present authors, CTCs were found to be associated with shorter recurrence-free survival and overall survival in patients with colorectal cancer with peritoneal metastases who had been treated with CRS and HIPEC. Patients in which CTCs were detected after neoadjuvant treatment had a far worse overall survival and earlier recurrence [30].

So far, CTCs have not been studied in patients with appendiceal neoplasms. A search in PubMed only revealed a recent published abstract as a systematic review for patients with appendiceal neoplasms and the role of circulating tumor DNA [31]. This abstract concluded that ctDNA was observed in more poorly differentiated appendiceal cancers. Patients with peritoneal metastases often respond poorly to current treatment [15]. An improved understanding of the molecular biology and the microenvironment of the tumor may provide insights into how we can devise more effective treatment. Hopefully, CTC-based assays may, in the future, give us important prognostic and predictive information [32]. 

Clinical evidence demonstrates that different cancer forms preferentially spread to certain anatomical locations. The biological mechanisms that determine lymphatic, hematogenous, and/or peritoneal spread are poorly understood [16,33]. Interestingly, our study identified a group of patients with appendiceal neoplasms in whom CTCs were present. Differences in molecular biology between, for example, colorectal cancer and appendiceal neoplasms—and even within the heterogenous group of epithelial appendiceal neoplasms—are likely to play an important role in CTC recovery. If clinical response could be predicted based on the presence of CTCs before objective validation in radiological imaging, CTCs could play an important role in guiding treatment. 

Our analyses were performed with the CellMate^®^ system, which is a novel platform technology that can identify CTCs. This system uses a microfluidic device to identify different cells based on biomechanical properties, including flowrate, biomechanical properties, surface interactions, and plasticity [20,21]. The expectation is that a broad sampling of CTCs would be obtained. Subsequently, the cells to be counted or analyzed for biomarkers would be selected among the initial broad sampling. In this smaller cohort, there was no obvious prognostic value to the CTCs, which is also a reasonable outcome. These tumors rarely produce hematogenous metastases. It would seem that even if CTCs make their way to the bloodstream, they may not possess the necessary cellular properties to establish systemic metastases in at least some stages. This is in concordance with well-known clinical findings: particularly, low-grade appendiceal neoplasms do not develop systemic metastases. Other reasons for the lack of prognostic impact may be the sample size, which is rather small to show prognostic benefit, particularly in a disease with such a low malignant potential. While prognostication may not have a future for CTCs in appendiceal neoplasms, it could potentially have a diagnostic one. Due to the surprisingly significant proportion of patients with CTCs in this study, there might be, for example, a role in determining the grading of the tumor preoperatively by using CTC diagnostics. One potential direction for a future study would be to try to broaden the sampling of CTCs analyzed by optimizing the platform technology, and another may be to determine whether the CTCs found in the blood can be used to determine the grading of the disease. Ultimately, genomic DNA sequencing of the individual cells will be required.

This study has several limitations. One limitation of the study is the inherent nature of the disease under scrutiny. Since appendiceal neoplasms with peritoneal metastases are extremely rare, the size of the study cohort—despite being a tertiary center with a catchment area of several millions—remains limited. Thus, this study could be seen either as a hypothesis-generating study or a pilot study assessing the presence of CTCs and their clinical importance in patients with appendiceal neoplasms. Although no correlation was found between the presence of CTCs in the bloodstream and early recurrence or overall survival, the mere presence of CTCs in a subgroup of patients may indicate that this group has a slightly different molecular subtype of appendiceal neoplasm. To further enhance our understanding of the role of CTCs in patients with epithelial appendiceal neoplasms, a larger multicenter study is needed. In the meantime, further subtyping of the groups of patients with CTCs present may shed light on the complex molecular nature of the disease. To include testing the biology of the cancer cells and not just the number of cells, the simplest hypothesis for future studies would be to consider what classic studies demonstrate: that it is the pattern of mutations that are the root cause of cancer. Therefore, the next logical step, which has until recently been technically challenging to take, is to study how genomic biomarkers correlate with local invasiveness, with hematological, lymphatic, and ascites-mediated metastatic spread, with synchronous or metachronous metastases, and with clinical outcomes. The results of such studies, in addition to the studies of cell numbers, would hopefully yield information on how the cancer-related mutations at the cellular level relate, stepwise, to the mechanisms of spread and ultimately to clinical outcomes. 

## 5. Conclusions

This is the first study to examine the presence, diagnostic potential, and clinical importance of CTC sampling in patients with epithelial appendiceal neoplasms. Our study shows that a subgroup of patients with this disease has detectable circulating tumor cells in their blood. Although we could not in our material show a difference in either tumor recurrence or overall survival between the groups (CTC-negative versus CTC-positive), it is interesting to compare our results with those of somatic genome sequencing data of solid tumor biopsy material that also do not allow, based on genomic profile alone, to predict future metastatic trajectory [34]. We believe that further single cell genomic DNA subtyping of the CTCs may potentially aid in the preoperative diagnostics of respective histopathological grading by singling out the somatic genomes present in the spreading of cancer cells. If successful, this could enhance the patient selection process preoperatively and management post-operatively. In conclusion, larger studies are needed to further clarify the role of CTC analysis as a prognostic biomarker in appendiceal epithelial neoplasms.

## Figures and Tables

**Figure 1 cancers-16-02441-f001:**
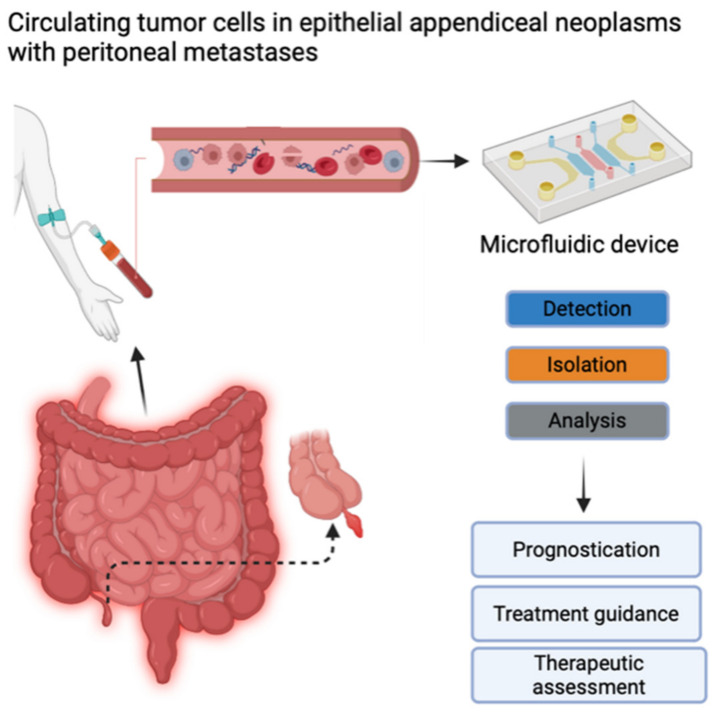
Circulating tumor cells (CTCs) contain important information about the molecular characteristics relevant to tumor progression and therapy. Potential clinical implications of measuring CTCs include prognostication, treatment guidance, and therapeutic assessment. Created with BioRender.com (accessed on 28 June 2024). Published with their permission.

**Figure 2 cancers-16-02441-f002:**
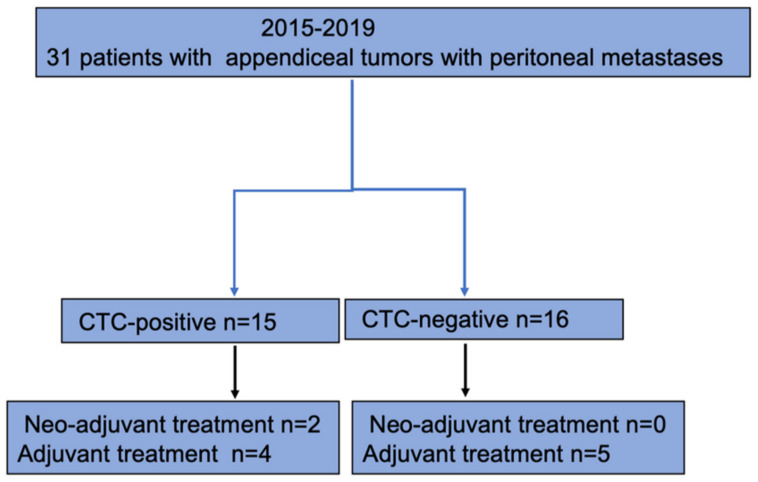
Flowchart of study population.

**Figure 3 cancers-16-02441-f003:**
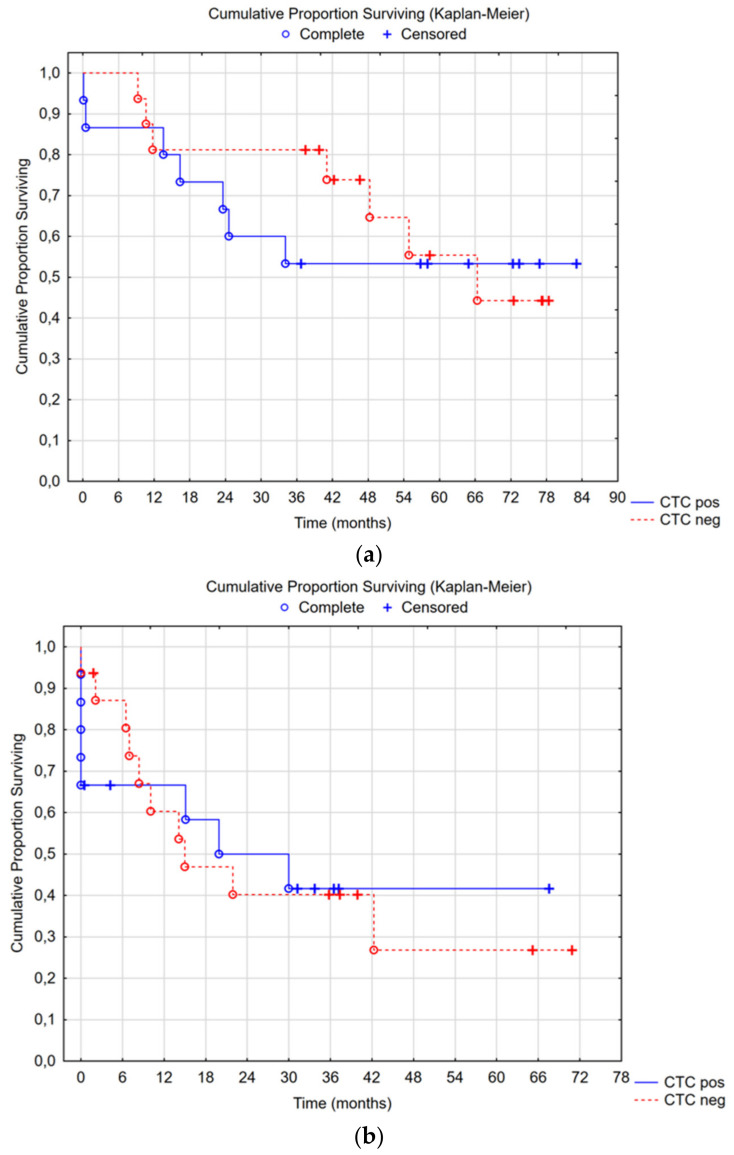
(**a**) Median overall survival in the CTC-positive group was not reached, with a 5-year survival rate of 53%; in the CTC-negative group, the median overall survival was 66 months (95% CI 34—not reached) with a 5-year survival rate of 55%, Cox’s F-test *p* = 0.355. (**b**) Median recurrence-free survival in the CTC-positive group was 20 months (95% CI 8—not reached), and in the CTC-negative group, it was 15 months (95% CI 8—not reached), Cox’s F-test *p* = 0.383.

**Table 1 cancers-16-02441-t001:** Baseline characteristics.

	No. Patients ^a^ N = 31	CTC-Positive N = 15	CTC-Negative N = 16	*p* Value
**Age (years) ^b^**	58 (54–67)	62 (51–68)	57 (55–67)	0.882
**Gender**				0.479
Men	17 (53)	7 (53)	10 (59)	
Women	14 (47)	8 (47)	6 (41)	
**BMI (kg/m^2^) ^b^**	26 (22–29)	26 (21–29)	26 (22–29)	0.970
**Karnofsky grade**				
100	22 (71)	10 (67)	12 (71)	0.651
90	4 (13)	2 (13)	2 (12)	
70	1 (4)		1 (5)	
Unspecified	4 (13)	3 (20)	2 (12)	
**CEA ^b^**	5 (2–66)	4 (2–63)	9 (2–75)	0.935
**CA19-9 ^b^**	26 (13–196)	27 (21–176)	15 (12–319)	0.713
**CA 125 ^b^**	38 (13–100)	47 (13–154)	35 (14–68)	0.683
**Primary histopathology**				
LAMN	15 (48)	9 (60)	6 (38)	0.376
Adenocarcinoma	11 (36)	4 (27)	7 (44)	
Goblet cell carcinoma	2 (7)	0	2 (12)	
Missing	3 (10)	2 (13)	1 (6)	
**Peritoneal Histopathology**				
Acellular mucin	2 (7)	1	1	0.755
Goblet cell	1 (3)	0	1	
MCP grade 1	6 (19)	3	3	
MCP grade 2	17 (55)	7	10	
MCP grade 3	2 (7)	2	0	
No positive histopathology	3 (10)	2	1	
**Low-grade tumors**	11 (36)	6	5	
**High-grade tumors**	20 (64)	9	11	0.716

BMI—body mass index; CEA—carcinoembryonic antigen; CA19-9—carbohydrate antigen 199; CA 125—cancer antigen 125; LAMN—low-grade appendiceal mucinous neoplasm; MCP—mucinous carcinoma peritonei. ^a^ With percentages in parentheses unless indicated otherwise; ^b^ values are median (i.q.r.).

**Table 2 cancers-16-02441-t002:** Surgical results and morbidity.

Type of Surgery	No. Patients ^a^ N = 31	CTC-Positive N = 15	CTC-Negative N = 16	*p* Value
CRS	2 (6)	1	1	0.168
CRS + HIPEC	22 (71)	9	13	
Debulking	3 (10)	1	2	
Open close	4 (13)	4	0	
**PCI score ^b^**	25 (4–35)	26 (5–35)	24 (4–34)	0.880
**HIPEC chemotherapy**				0.577
Mitomycin C	14 (45)	6	8	
Oxaliplatin	8 (26)	3	5	
Not specified	1 (3)	1	1	
Not given	8 (26)	5	3	
**Operation time ^b^**	335 (169–545)	304 (99–411)	353 (212–575)	0.112
**CC-Score**				0.168
CCS-0	15 (48)	8	7	
CCS-1	9 (29)	2	7	
CCS-2	0	0	0	
CCS-3/open-and-close	7 (23)	5	2	
**Clavien-Dindo**				0.820
1	4 (13)	2	2	
2	19 (61)	9	10	
3a	1 (3)	0	1	
3b	3 (10)	1	2	
5	2 (7)	0	2	
Not specified	2 (7)	1	1	
**Complications**				0.364
Wound dehiscence	1 (3)	0	1	
Intraabdominal bleeding	1 (3)	0	1	
Bowel perforation	1 (3)	1	0	
Multiorgan failure	1 (3)	1	0	
Pulmonary embolism	1 (3)	1	0	
**Neoadjuvant chemotherapy**	2 (7)	2	0	0.226
**Adjuvant chemotherapy**	9 (29)	4	5	0.647

CRS—cytoreductive surgery; HIPEC—hyperthermic intraperitoneal chemotherapy; PCI—peritoneal cancer index; CC-Score—complete cytoreduction score. ^a^ With percentages in parentheses unless indicated otherwise; ^b^ values are median (i.q.r.).

**Table 3 cancers-16-02441-t003:** Cox’s regression model and overall survival.

	Univariable HR, 95% CI	*p* Value	Multivariable HR, 95% CI	*p* Value
**Age (years)**				
<70	Reference		Reference	
≥70	1.43 (0.32–6.43)	0.640	4.13 (0.71–24.10)	0.115
**Gender**				
Women	Reference		Reference	
Men	2.50 (0.78–8.01)	0.122	1.71 (0.52–5.59)	0.375
CTC-negative	Reference		Reference	
CTC-positive	1.18 (0.41–3.37)	0.758	1.32 (0.45–3.89)	0.612
PCI score				
<13	Reference		Reference	
>12	2.87 (1.51–5.44)	0.001	5.21 (1.16–23.38)	0.031
Low-grade tumors	Reference		Reference	
High-grade tumors	0.81 (0.29–2.45)	0.750	0.97 (0.32–2.89)	0.950

**Table 4 cancers-16-02441-t004:** Cox’s regression model and recurrence-free survival.

	Univariable HR, 95% CI	*p* Value	Multivariable HR, 95% CI	*p* Value
**Age (years)**				
<70	Reference		Reference	
≥70	1.24 (0.68–2.28)	0.484	0.72 (0.09–5.64)	0.754
**Gender**				
Women	Reference		Reference	
Men	1.10 (0.65–1.83)	0.742	1.11 (0.39–3.16)	0.841
CTC-negative	Reference		Reference	
CTC-positive	1.20 (0.71–2.03)	0.494	0.87 (0.33–2.29)	0.774
PCI score				
<13	Reference		Reference	
>12	2.54 (1.48–4.36)	<0.001	5.17 (1.60–16.63)	0.006
Low-grade tumors	Reference		Reference	
High-grade tumors	2.65 (0.850–8.24)	0.093	3.97 (1.18–13.40)	0.026

## Data Availability

The data may be available from the corresponding author based on reasonable request.

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
