# Peer review of "Clinical Significance of Circulating Tumor Cells in Epithelial Appendiceal Neoplasms with Peritoneal Metastases"

_cancers, 2024, doi:10.3390/cancers16132441_

Round 1

Reviewer 1 Report

Comments and Suggestions for Authors

The manuscript presents a valuable study investigating the prevalence of circulating tumour cells (CTCs) and their association with clinical and pathological factors in patients with appendiceal tumours. The authors have made a commendable effort to fill a gap in the current medical literature by studying CTCs in this context for the first time.

The lack of association between CTC positivity and survival or PCI score may indicate the need for further studies to understand the role of CTCs in this patient population.

The unexpected finding of CTCs in almost half of the patients warrants a deeper investigation of the biological significance of these cells in appendiceal tumours.

The manuscript could use a stronger discussion of the possible reasons why CTCs do not correlate with oncological outcomes, which could help in future studies.

Comments:

1.     The sample size is too small to use the Chi-square test; I recommend using the two-sided Fisher's exact test instead. Moreover, the Chi-square test is not used when one of the compared values equals 0.

2.     How is the effectiveness of the therapy assessed in these patients?

3.     The abbreviations DPAM, disseminated peritoneal adenomucinosis; PMCA, peritoneal mucinous carcinomatosis; MCP, mucinous carcinoma peritonei (new classification for pseudomyxoma peritonei), are noted in the footnote to Table 1, although they are not used in the table itself.

4.     Were there any associations between the clinical-pathological parameters of the patients and Time to recurrence and overall survival?

5.     How similar in aggressiveness are the histological subtypes of tumors included in the study?

6.     Was ascites encountered in these patients?

7.     What was done with the two blood samples? Were the values averaged?

Author Response

Answers to reviewers CTC in appendiceal neoplasms

The manuscript presents a valuable study investigating the prevalence of circulating tumour cells (CTCs) and their association with clinical and pathological factors in patients with appendiceal tumours. The authors have made a commendable effort to fill a gap in the current medical literature by studying CTCs in this context for the first time.

The lack of association between CTC positivity and survival or PCI score may indicate the need for further studies to understand the role of CTCs in this patient population.

The unexpected finding of CTCs in almost half of the patients warrants a deeper investigation of the biological significance of these cells in appendiceal tumours.

The manuscript could use a stronger discussion of the possible reasons why CTCs do not correlate with oncological outcomes, which could help in future studies.

Answer:

We have added some more information in the discussion concerning the reason why CTC do not correlate with oncological outcomes. The main reason which we cited already is the appendiceal low grade tumors basically never metastasize systemically. This is too a great degree a locoregional abdominal disease. In this respect, it is logical that systemic CTCs do not seem to affect the prognosis. We have also added another reason which is that the study is rather small considering that appendiceal tumors are of low malignant potential (in particular the low-grade tumors). More than half the patients are alive after 10 years; this may also play an important role in why we do not see any prognostic benefit.

Furthermore, classic studies with model organisms demonstrate that the intravasated cancer cells (CTCs) are the precursors of metastases. Applying the principle of Occam’s razor, i.e that if all things are equal the simplest, and therefore preferred, hypothesis would be that the number of intravasated cancer cells would predict the number of metastases which in turn would predict the disease specific survival outcomes. Interestingly the above hypotheses are shown to be false as tested by the present study. Clearly reality is more complex than these simplest hypotheses would predict. Stated differently, the biology of the cells, not just the number of cells, matter.

To include testing the biology of the cancer cells, not just the number of cells, the simplest hypothesis for future studies would be to consider that other classic studies demonstrate that it is the pattern of mutations of the cancer cells that are the root cause of cancer. Therefore, the next logical step, that has until recently been technically challenging to take, is to study how genomic biomarkers correlate with local invasiveness, with hematological, lymphatic, and ascites mediated metastatic spread, with synchronous or metachronous metastases and with clinical outcomes. The results of such studies, in addition to the studies of cell numbers would be expected to yield information of how the cancer-related mutations at the cellular level stepwise relate to the mechanisms of spread and ultimately with clinical outcomes.

Comments:

  1. The sample size is too small to use the Chi-square test; I recommend using the two-sided Fisher's exact test instead. Moreover, the Chi-square test is not used when one of the compared values equals 0.

Answer:

Thank you for pointing this out. We have amended this in the manuscript.

  1. How is the effectiveness of the therapy assessed in these patients?

Answer:

Thank you for your comment. The study per se is not assessing the effectiveness of the therapy. For this to be a realistic goal the study design has to be a RCT which combines assessment of two different treatments with specified endpoints, in combination with the measurement of CTCs present in the bloodstream. Our study should be seen as an exploratory hypothesis-generating study that examines the presence or non-presence of CTCs in the bloodstream.

  1. The abbreviations DPAM, disseminated peritoneal adenomucinosis; PMCA, peritoneal mucinous carcinomatosis; MCP, mucinous carcinoma peritonei (new classification for pseudomyxoma peritonei), are noted in the footnote to Table 1, although they are not used in the table itself.

Answer:

Thank you for noticing this, we have deleted the unnecessary acronyms.

  1. Were there any associations between the clinical-pathological parameters of the patients and Time to recurrence and overall survival?

Answer:

Thank you for bringing this up. In a multivariable Cox regression analysis, there were no correlation between age >70 years old, sex, whether CTCs-positive or CTC-negative or low versus high-grade tumors and overall survival. There was a strong correlation between PCI score of 13 or above and overall survival (HR 5.21, 95% CI 1.16-23.38, p=0.031). In a multivariable Cox regression analysis, there was a correlation between PCI score of 13 and above (HR 5.17, 95% CI 1.60-16.63, p=0.006), and high-grade tumors (HR 3.97, 95% CI 1.18-13.40, p=0.026) and recurrence-free survival. There was no association between CTC-positivity and earlier recurrence of the disease. For clarification, this information has been added in the manuscript, see page 7, and Tables 3 and 4.

  1. How similar in aggressiveness are the histological subtypes of tumors included in the study?

Answer:

Thank you for asking this question. As mentioned above, the low-grade tumors have very good prognosis with more than 75% of patients alive after 10 years. Likewise, the high-grade tumors also have a good prognosis but with greater heterogeneity - goblet cell tumors and signet-ring cell tumors in the high-grade tumor have clearly poor prognosis. The rest of the high-grade tumors (mucinous carcinoma peritonei grade 2) have more the 50% alive after 10 years. The general good prognosis in appendiceal tumors may be one of the reasons that the prognostic benefit is not obvious in the pilot study. Larger future studies are needed.

  1. Was ascites encountered in these patients?

Answer:

No classic serous ascites was found in these patients. However, these tumors are in general very mucinous and as such may have large mucinous ascites or mucinous implants. Radiologically, it will look like mucinous ascites.

  1. What was done with the two blood samples? Were the values averaged?

Answer:

In order to more easily perform the statistical analyses, patients were categorized as having CTC present vs not present. The two blood samples were combined with the absolute number of CTCs added together as well as the total blood volume added together. Had we found a span of different CTC numbers, it would have been interesting to look at the number of CTC per volume blood. Unfortunately, usually less than 10 and quite often just a few CTCs were captured. There was not enough power to look at different CTC levels. We have opted to put all patients with perioperative CTCs into one group regardless of the number of CTCs. The CTC negative group was negative in both samples, i.e. no CTCs found at all.

Reviewer 2 Report

Comments and Suggestions for Authors

This is an interesting work submitted to Cancers that is adding significance to the field of circulating tumor cells (CTC) in neoplasms, specifically analyzing their role in the appendiceal neoplasm with peritoneal metastases. This argument can have a large impact on early detection of tumor and consequentially early treatment. Methods are well described even if it could be better to summarize all of them in only one period to a better comprehension of the processes. Figure are well edited and clarify, so I confirm your idea to put them into the manuscript. I suggest to be careful on the English which is in some part of the manuscript too scholar and not so exhausting in confront of the argument.  The number of patients enrolled is too low to reach a statistical positive event, but I think that it is the better enlistment possible. Finally I think that it is a well done work with excellent ideas which can lead also other branches of oncological medicine. However, in the attempt to have a comparison with the use of CTC in other oncological areas I suggest to add DOI: 10.1634/theoncologist.2018-0784 and DOI: 10.1016/j.clgc.2017.01.011 which are clear example of that.In conclusion, the article is a great effort from the authors, yet it is missing some major revisions to be further processed. I do please remind the authors to bold and track the changes in their manuscript so as to facilitate the review process.

Comments on the Quality of English Language

This is an interesting work submitted to Cancers that is adding significance to the field of circulating tumor cells (CTC) in neoplasms, specifically analyzing their role in the appendiceal neoplasm with peritoneal metastases. This argument can have a large impact on early detection of tumor and consequentially early treatment. Methods are well described even if it could be better to summarize all of them in only one period to a better comprehension of the processes. Figure are well edited and clarify, so I confirm your idea to put them into the manuscript. I suggest to be careful on the English which is in some part of the manuscript too scholar and not so exhausting in confront of the argument.  The number of patients enrolled is too low to reach a statistical positive event, but I think that it is the better enlistment possible. Finally I think that it is a well done work with excellent ideas which can lead also other branches of oncological medicine. However, in the attempt to have a comparison with the use of CTC in other oncological areas I suggest to add DOI: 10.1634/theoncologist.2018-0784 and DOI: 10.1016/j.clgc.2017.01.011 which are clear example of that.In conclusion, the article is a great effort from the authors, yet it is missing some major revisions to be further processed. I do please remind the authors to bold and track the changes in their manuscript so as to facilitate the review process.

Author Response

Reviewer 2:

This is an interesting work submitted to Cancers that is adding significance to the field of circulating tumor cells (CTC) in neoplasms, specifically analyzing their role in the appendiceal neoplasm with peritoneal metastases. This argument can have a large impact on early detection of tumor and consequentially early treatment. Methods are well described even if it could be better to summarize all of them in only one period to a better comprehension of the processes.

Answer:

The methods section is in many ways the most important part of a work, since it should be transparent so as to make it possible to follow the different steps in the process.  We have reviewed our methods section and are not so sure what the author would like us to change. We have kept the methods short and concise, not even a full page. We are open to making changes to the methods section, but in that case please provide some more details as to what is unclear and we will happily try to clarify the methods section more.

Figure are well edited and clarify, so I confirm your idea to put them into the manuscript. I suggest to be careful on the English which is in some part of the manuscript too scholar and not so exhausting in confront of the argument.  The number of patients enrolled is too low to reach a statistical positive event, but I think that it is the better enlistment possible.

Answer:

We agree with the reviewer that a larger number of patients included in the study would have been desirable. For this to be feasible muti-center studies are required, since the diagnosis is uncommon. Since this prospective study was a hypothesis-generating study, and CTCs had not previously been studied in patients with appendiceal neoplasm, a power analysis was not performed.

Finally I think that it is a well done work with excellent ideas which can lead also other branches of oncological medicine. However, in the attempt to have a comparison with the use of CTC in other oncological areas I suggest to add DOI: 10.1634/theoncologist.2018-0784 and DOI: 10.1016/j.clgc.2017.01.011 which are clear example of that.In conclusion, the article is a great effort from the authors, yet it is missing some major revisions to be further processed. I do please remind the authors to bold and track the changes in their manuscript so as to facilitate the review process.

Answer:

We thank the reviewer for these two interesting papers. We have read them, and included their conclusions in our own discussion (pages 8-9). We have made changes throughout according to both reviewer comments in order to correct possible deficits in the manuscript.

Thank you for the opportunity to respond to these comments.

Round 2

Reviewer 1 Report

Comments and Suggestions for Authors

I am grateful for the authors' comprehensive responses to my inquiries. Their clarifications have effectively addressed the concerns I had, enhancing my comprehension of the study's broader impact. The manuscript now clearly demonstrates its significant contribution to the field, underscored by the extensive revisions. It offers insightful advancements that enrich our collective understanding.

Reviewer 2 Report

Comments and Suggestions for Authors

I thank you for your response